# Genetic adaptation to amoxicillin in *Escherichia coli*: The limited role of *dinB* and *katE*

**Lisa Teichmann[1], Marcus Wenne[2,3], Sam Luitwieler[1], Gaurav Dugar[1], Johan Bengtsson-Palme[2,3,4], Benno ter Kuile[1]***

1 Swammerdam Institute of Life Sciences, Molecular Biology and Microbial Food Safety, University of Amsterdam, Amsterdam, The Netherlands, 2 Department of Life Sciences, Division of Systems and Synthetic Biology, SciLifeLab, Chalmers University of Technology, Gothenburg, Sweden, 3 Centre for Antibiotic Resistance Research (CARe) in Gothenburg, Gothenburg, Sweden, 4 Department of Infectious Diseases, Institute of Biomedicine, University of Gothenburg, Gothenburg, Sweden

* b.h.terkuile@uva.nl

**Data Availability Statement:** All relevant data can be found in the following GitHub link: https://github.com/mnyt-aqw/Teichmann_2024_Amoxicillin.

## Abstract

Bacteria can quickly adapt to sub-lethal concentrations of antibiotics. Several stress and DNA repair genes contribute to this adaptation process. However, the pathways leading to adaptation by acquisition of *de novo* mutations remain poorly understood. This study explored the roles of DNA polymerase IV (*dinB*) and catalase HP2 (*katE*) in *E. coli*'s adaptation to amoxicillin. These genes are thought to play essential roles in beta-lactam resistance—*dinB* in increasing mutation rates and *katE* in managing oxidative stress. By comparing the adaptation rates, transcriptomic profiles, and genetic changes of wild-type and knockout strains, we aimed to clarify the contributions of these genes to beta-lactam resistance. While all strains exhibited similar adaptation rates and mutations in the *frdD* gene and *ampC* operon, several unique mutations were acquired in the Δ*katE* and Δ*dinB* strains. Overall, this study distinguishes the contributions of general stress-related genes on the one hand, and *dinB*, and *katE* on the other hand, in development of beta-lactam resistance.

## 1. Introduction

Since bacterial resistance mechanisms were first described, considerable efforts have been devoted to revealing the underlying molecular machinery involved in developing antimicrobial resistance. Understanding the cellular pathways involved is essential for addressing the growing threat looming over modern medicine–an era where previously effective antibiotics are increasingly unable to treat resistant infections (https://www.who.int/publications/i/item/no-time-to-wait-securing-the-future-from-drug-resistant-infections). Over billions of years, bacteria have evolved distinct and sophisticated defence mechanisms that not only enable survival but often allow them to thrive under adverse conditions. Many of these complex defence systems are versatile, protecting against multiple stressors and effectively pre-adapting bacteria

**Funding:** This study was financed by The Netherlands Food and Consumer Product Safety Authority.

**Competing interests:** The authors have declared that no competing interests exist.

to various forms of harm [1–3]. Environmental stressors, such as starvation, have been shown to activate mutagenic pathways, which, in turn, contribute to the acquisition of antibiotic resistance [4–6]. This connection underscores the complex interplay between bacterial survival strategies and the emergence of resistance, highlighting the importance of an integrative approach when investigating these adaptive mechanisms.

Exposure to beta-lactam antibiotics such as amoxicillin is known to trigger a range of stress responses in bacterial cells, including the SOS and oxidative stress response [7,8]. Amoxicillin, with its broad-spectrum activity and relative safety, holds a central position in human healthcare. It is crucial for treating a wide array of bacterial infections, from common respiratory and urinary tract infections to pneumonia and skin infections [9]. The extensive use of amoxicillin underscores the importance of understanding how bacterial adaptation to this antibiotic occurs.

DNA polymerase IV, encoded by the *dinB* gene, plays a vital role in the SOS response [10]. The *dinB* gene facilitates error-prone DNA synthesis, introducing mutations that, while potentially harmful, can also allow bacteria to adapt and survive under challenging conditions [11–15]. The role of *dinB* in beta-lactam resistance is complex. Although beta-lactam antibiotics do not directly damage DNA, they have been shown to induce a two-component signal transduction system that activates the SOS response [7]. Moreover, *dinB* has been implicated in ampicillin-induced mutagenesis in *Escherichia coli* and other bacterial species and appears to be responsible for many of the mutations that arise following beta-lactam exposure [16]. Interestingly, these mutations occur independently of the typical SOS regulon and RpoS sigma factor activation, suggesting that other, yet-to-be-identified mechanisms might be involved [17]. One possible link could be the role of intracellular reactive oxygen species (ROS), which have been shown to increase following sub-inhibitory exposure to various bactericidal antibiotics, including aminoglycosides and beta-lactams like ampicillin [13,18,19].

This increase in ROS is hypothesized to result from increased metabolic rates, causing perturbations of the TCA cycle [20,21], potentially leading to an increase in mutations. On the one hand, these mutations may contribute to antibiotic resistance. On the other hand, they could eventually function as a secondary killing mechanism in addition to the antimicrobial properties of the antibiotics [8,22]. However, this 'ROS theory' remains a subject of debate and the ROS-induced lethal effect of bactericidal antibiotics seems to depend on antibiotic concentrations [23,24]. While it is well-established that bactericidal antibiotics elevate intracellular ROS levels, the exact role of ROS in antibiotic-induced mutagenesis and adaptive mutations is still not fully understood.

In this context, the enzyme KatE (catalase HP2) plays a crucial role in *E. coli*'s defence against oxidative stress. Catalase enzymes, such as KatE, are essential for detoxifying ROS by breaking down hydrogen peroxide ($H_2O_2$) into water and molecular oxygen [25]. This detoxification process significantly enhances bacterial fitness and resilience, particularly under conditions where antibiotic exposure induces ROS production [26]. KatE is induced by the stringent response in *E. coli*, which is activated under environmental stress conditions such as nutrient starvation [26]. The stringent response leads to the downregulation of cell growth-related transcription and the upregulation of survival mechanisms [27]. Interestingly, *E. coli* in a stringent state shows increased tolerance to many antibiotic classes [28,29]. Elevated levels of KatE have been proposed as a contributing factor to this increased tolerance, as a correlation between catalase activity and antibiotic tolerance has been observed during the stringent response [26]. However, the precise cause of this association remains unclear. A likely explanation could be that the catalase reduces the intracellular ROS levels induced by the antibiotic enough to enable survival [26].

In this study, we investigated the genetic and transcriptomic alterations in *E.coli* MG1655, a well-established model organism, and two corresponding knockout mutants (Δ*dinB*, Δ*katE*) when exposed to sub-lethal concentrations of amoxicillin. We initially hypothesized that the absence of *dinB* would result in a reduced mutation rate, hindering *E. coli*'s ability to adapt to these conditions. Meanwhile, the deletion of *katE* was expected to lead to difficulties in adaptation due to increased oxidative stress, although it might also cause an elevated mutation rate which could elevate adaptation. This would impair *E. coli*'s ability to adapt to these conditions, given the previously reported lower mutation rates in *dinB* knockouts and the increased oxidative stress observed in *katE*-deficient strains [30,31]. Contrary to our expectations, the knockout strains adapted at a similar rate to the wild-type strain. This suggests that *dinB* and *katE* may not be as critical for beta-lactam adaptation as generally assumed.

Consistent mutations in the *ampC* operon were found across all three strains, highlighting its importance in amoxicillin resistance acquisition. Additionally, other genes were frequently mutated, with some identical mutations observed across different strains and replicates, including a notable mutation in *rpoD*, suggesting a significant role of this sigma factor under antibiotic stress.

Furthermore, the role of *frdD* mutations and regulation seemed to be more multifaceted than previously assumed, indicating potential involvement in resistance mechanisms beyond *ampC* regulation. The consistent changes observed in the transcriptomic regulation of the toxin/antitoxin system *prlF/yhaV* further suggests a role of this system in the acquisition of resistance.

## 2. Material and methods

### Strains, growth conditions, and antimicrobial agents

The *Escherichia coli* strain MG1655 was utilized as the wild-type strain. Single gene knockout mutants JW0221 (Δ*din*B749::kan) and JW1721 (Δ*katE*731::kan) were obtained from the KEIO collection, supplied by Horizon Discovery Ltd. These knockout mutants contained kanamycin-resistant cassettes flanked by FLP recognition target (FRT) sites, which were removed using the pCP20 method prior to the experiments [32,33].

Bacterial cultures were grown in lysogeny broth (LB) containing 10 g/L NaCl, either in liquid or solid form. All strains were initially cultured to an OD600 of 0.1 and incubated at 37°C with shaking at 200 rpm overnight. For longer weekend incubations, the starting OD600 was reduced to 0.01, and the incubation temperature was adjusted to 30°C. Amoxicillin was supplied by Merck KGaA. A stock solution (10 mM) was prepared which was stored at 4°C and used within three days of preparation and filter sterilization.

### Minimum inhibitory concentrations (MIC)

Minimum inhibitory concentrations (MICs) were measured twice a week in duplicate for each strain using the broth microdilution method [34]. Readings were taken every 10 minutes, with 5 minutes shaking intervals between measurements.

### Evolution experiment

Evolution experiments were conducted as described previously [35]. At the start of the experiment, the MIC of each strain was determined. Amoxicillin exposure began at a concentration of 2 μg/mL, equivalent to ¼ of the MIC for all strains. Following overnight incubation, the OD600 was measured. If the OD600 was above 65% of the OD600 of the previous culture, the amoxicillin concentration was doubled in fresh medium. If the OD600 was below this

threshold, the culture was transferred to fresh medium with the same amoxicillin concentration as the previous day. In parallel, each bacterial strain was cultured without antibiotic exposure as a biological control. Three technical replicates were performed. The experiment was terminated when the culture reached and tolerated an amoxicillin concentration of 1024 μg/mL for multiple days. If the culture could not adapt to this concentration, the experiment was stopped after 10 transfers without successful increase to a higher antibiotic concentration.

## DNA isolation and whole genome sequencing

Genomic DNA was isolated from each culture at both the start and end of the evolution experiment using the PureLink Genomic DNA Mini Kit (Thermo Fisher Scientific), with some modifications to the manufacturer's protocol. Cultures were pelleted by centrifugation at 12,000 × g for 1 minute, and the resulting pellet was resuspended in 300 μL of TE buffer. Then 40 μL of 10% SDS, 3 μL of 0.5 M EDTA, and 20 μL of proteinase K were added. Cell lysis was achieved by incubating the samples at 65°C for 5 minutes in a heating block. Following lysis, 20 μL of RNase A was added, and the samples were incubated at room temperature for 3 minutes. After RNase addition, the protocol was followed as recommended by the supplier.

The quality and quantity of the isolated genomic DNA were assessed using a NanoDrop spectrophotometer (Thermo Fisher Scientific) to measure absorbance at 260/280 nm. The integrity of the DNA was further confirmed by running the samples on a 1% agarose gel. The purified genomic DNA was then used for library preparation according to the manufacturer's protocol for the NEBNext Ultra II FS DNA Library Prep Kit for Illumina (New England BioLabs). The libraries were subsequently sequenced using the Illumina sequencing platform.

## RNA isolation and sequencing

Frozen cultures were thawed on ice, followed by overnight inoculation in LB with the appropriate antibiotic concentration (S2 Table). Total RNA was then extracted using PCI for RNA isolation. Ethanol precipitation was used to purify the RNA, and any residual genomic DNA was removed with DNase I (New England BioLabs) digestion according to the manufacturer's instructions. RNA integrity was confirmed by visualizing the integrity of 16S and 23S ribosomal RNA bands on an 1% agarose gel. The NEBNext rRNA Depletion Kit (New England BioLabs) was employed to deplete ribosomal RNA, as per the manufacturer's protocol. The rRNA-depleted RNA was then prepared for sequencing using the NEBNext Ultra II Directional RNA Library Prep Kit for Illumina (New England BioLabs) and sequenced on an Illumina NextSeq 550 platform. Agencourt Ampure XP (Beckman Coulter) magnetic beads were used during library preparation.

## Transcriptome analysis

The entire transcriptomics workflow was designed as a Nextflow pipeline v21.10.6 [36]. Single ended RNA-Seq data was quality controlled and trimmed using TrimGalore! (https://github.com/FelixKrueger/TrimGalore?tab=readme-ov-file) v0.6.7 with settings—phred33 -e 0.1—quality 28. Afterwards, MultiQC v1.13 [37] was used to create a combined quality report for all samples. Next, the reads that passed quality control for all samples were mapped against the MG1655 reference genome (RefSeq: GCF_000005845.2) using Bowtie2 v2.3.5.1 and the resulting bam files were sorted using Samtools v1.3.1. Finally, the sorted bam files were converted to transcript counts using featureCounts v2.0.1 with default settings in single-end mode [38]. The differential gene expression analysis, GO term and KEGG pathway analysis was performed using edgeR v4.0.16. A gene was considered differentially expressed if the FDR corrected p-value was equal or below 0.05 [39]. A different cutoff for the GO and KEGG pathway

analysis was applied following recommendations by the authors of the goana and kegga packages, instead using a p-value of 0.001.

### Genome analysis

Like the transcriptomic workflow, the genome analysis was also performed using Nextflow. Pair-end genomic reads were quality controlled, and adapters were removed the same way as for the RNA-Seq data. To identify mutations, as well as their frequency in the bacterial population, Breseq v0.37.1 [40] was used with the—polymorphism-prediction flag and supplied with either the MG1655 reference genome: NC_000913.3 or the corresponding reference genome for the knock out strains: NZ_CP009273.1. To analyze gene copy number the paired trimmed reads were mapped against the MG1655 reference genome using Bowtie2. The Samtools view function (v1.3.1) was used to convert the sam file to a binary bam file [41]. Potential PCR duplicates were removed using the Samtools rmdup function with the -S flag (pair-end mode). Finally, the positions of the mapped reads were extracted using the Samtools view function, in combination with a Perl script (v5.32.1). The CNOGpro R library (v1.1) was used to convert the mapping data to copy numbers using the CNOGpro function with windowlength = 100, followed by the functions normalizeGC and runBootstrap with settings 'replicates = 1000, quantiles = c(0.025, 0.975)' [42], generating bootstrapped copy numbers for each gene.

### Data analysis

The data analysis was performed using Python v 3.11.8 or R v4.3.2 in combination with Jupyter Lab v 4.0.11 (https://github.com/jupyterlab/jupyterlab) and Quarto v1.4.553 (https://github.com/quarto-dev/quarto-cli). Finally, ggplot2 (https://ggplot2.tidyverse.org) and Seaborn were used for data visualization [43].

### Protein structure prediction

Protein structure changes resulting from the mutated genes were predicted using DDMut [44]. Initially, the protein sequence in PDB format was downloaded from UniProt (1SIG). Next, the identified mutation from the whole genome sequencing data was incorporated into this sequence. The DDMut online tool was then used to predict the new protein structure and stability (https://biosig.lab.uq.edu.au/ddmut/).

## 3. Results

### Deletion of *katE* or *dinB* does not alter antibiotic adaptation rates

We investigated the adaptation rate of the wild-type compared to the *dinB* and *katE* knockout strains and found no significant differences (Fig 1). Despite their limitations in stress responses considered crucial for beta-lactam resistance, their adaptation dynamics closely reflected that of the wild-type *E. coli*. Indeed, the biological and technical variations between replicates were comparable to the differences observed between the strains, suggesting that knocking out *dinB* and *katE* did not confer any notable disadvantage in adaptation to amoxicillin. All strains exhibited a uniform adaptation rate until the clinical resistance threshold of 8 µg/mL (https://mic.eucast.org) and above.

### Mutations in the *ampC* Operon and *rpoD* are associated with amoxicillin resistance

Whole genome sequencing of the adapted cultures revealed that the strains only gained a small set of partially overlapping and even identical mutations (see Fig 1 and Table 1). We focused

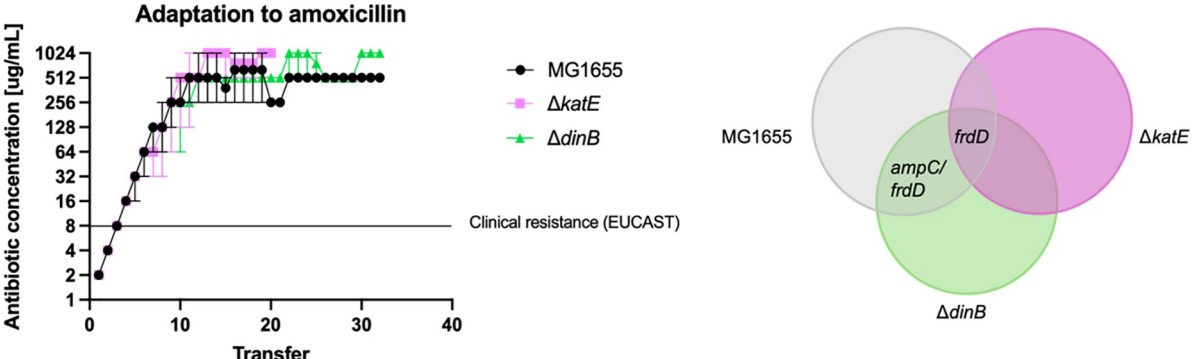

**Fig 1. Adaptation to amoxicillin.** Comparative analysis of the adaptation to amoxicillin across multiple transfers in three bacterial strains: MG1655 (black/grey), ΔkatE (purple), and ΔdinB (green) with three biological replicates for each strain. On the left: The y-axis represents the antibiotic concentration in micrograms per milliliter on a logarithmic scale, while the x-axis represents the number of transfers. The Venn diagram illustrates the overlapping mutations found in the three bacterial strains. Only mutations with >60% frequency and occurrence in at least 2 replicates are displayed. All strains shared a mutation in frdD. MG1655 and ΔdinB had a mutation between ampC/ frdD in common. More detailed information can be found in Table 1.

on mutations present in at least 60% of the population and only considered mutations that appeared in at least two out of the three replicates for each strain. This threshold was chosen based on the frequency distribution observed within the samples.

All three strains acquired a mutation in *frdD*, a gene within the *frd* operon which contains the promoter of the *ampC* beta-lactamase gene [45]. Mutations in this gene have been found to

**Table 1. Mutations acquired during amoxicillin exposure.**

| Frequency | Type | Sample | Gene Position | Gene | Effect | Genome position | Strand |
|---|---|---|---|---|---|---|---|
| 1 | SNP | ΔdinB | intergenic (-39/+24) | ampC/frdD | G→T | 4,368,800 | ← |
| 1 | DEL | ΔdinB | intergenic (-32/+29) | ampC/frdD | Δ3 bp | 4,368,793 | ← |
| 1 | DEL | MG1655 | intergenic (-32/+29) | ampC/frdD | Δ3 bp | 4,378,976 | ← |
| 1 | SNP | MG1655 | intergenic (-26/+37) | ampC/frdD | C→A | 4,378,970 | ← |
| 1 | SNP | MG1655 | intergenic (-26/+37) | ampC/frdD | C→G | 4,378,970 | ← |
| 1 | SNP | ΔdinB | 298 | frdD | V100L (GTT→CTT) | 4,368,886 | ← |
| 1 | SNP | ΔdinB | 353 | frdD | T118I (ACA→ATA) | 4,368,831 | ← |
| 1 | SNP | ΔkatE | 353 | frdD | T118I (ACA→ATA) | 4,368,831 | ← |
| 1 | INS | ΔkatE | coding (338/360 nt) | frdD | +A | 4,368,846:1 | ← |
| 1 | SNP | MG1655 | 298 | frdD | V100L (GTT→CTT) | 4,379,069 | ← |
| 1 | INS | MG1655 | coding (341/360 nt) | frdD | +C | 4,379,026:1 | ← |
| 1 | SNP | MG1655 | 332 | frdD | V111D (GTC→GAC) | 4,379,035 | ← |
| 1 | SNP | ΔdinB | 1334 | rpoD | D445V (GAT→GTT) | 3,207,739 | → |
| 0.07 | SNP | ΔdinB | 1334 | rpoD | D445V (GAT→GTT) | 3,207,739 | → |
| 1 | SNP | ΔkatE | 1334 | rpoD | D445V (GAT→GTT) | 3,207,739 | → |
| 1 | SNP | ΔkatE | 1334 | rpoD | D445V (GAT→GTT) | 3,207,739 | → |
| 1 | SNP | MG1655 | 1334 | rpoD | D445V (GAT→GTT) | 3,214,380 | → |

The table provides detailed information on the mutations identified in the three strains. It includes the frequency of each mutation, the type of mutation, the specific position within the gene, the effect of the mutation, at what position in the genome the change occurred as well on what strand the gene is located. The genome position is relative to each samples corresponding reference genome. NC_000913.3 for MG1655 and NZ_CP009273.1 for ΔdinB and ΔkatE. We observed that some mutations occurred at the same positions across different strains, highlighting the potential importance of these specific genetic changes in the adaptive response to amoxicillin.

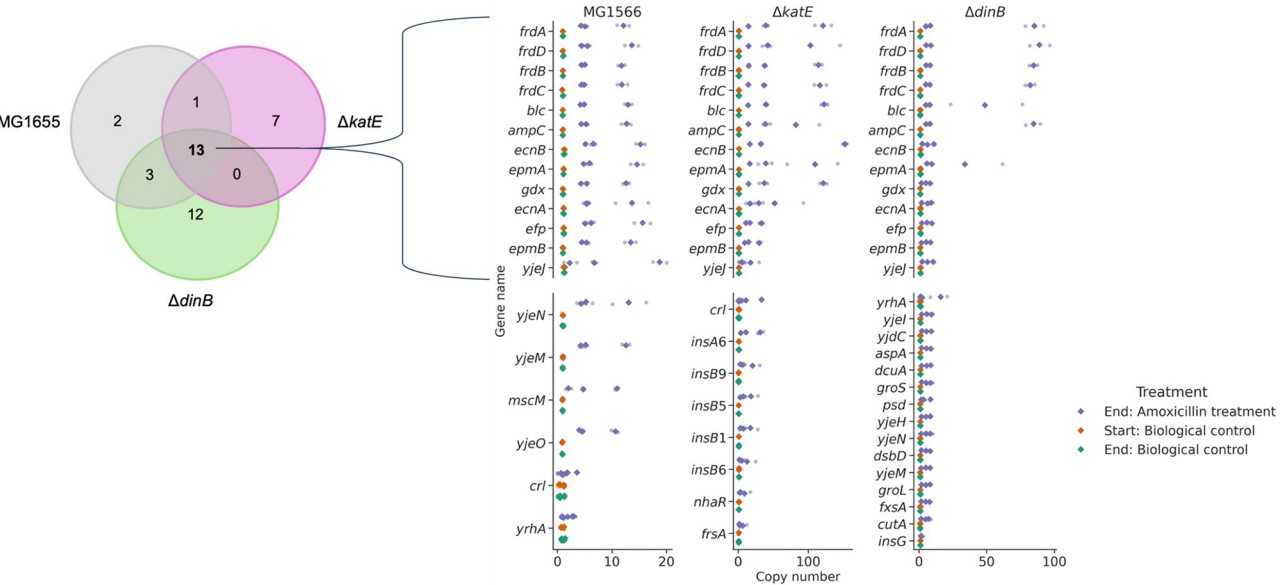

**Fig 2. Genes with a bootstrapped abundance above 1.5.** The graph illustrates the number of genes that are amplified ≥ 1.5 times. The rhomboid shape represents the bootstrapped values. The transparent circles represent the corresponding lower and upper bounds.

influence transcription levels of *ampC* [46–48]. Furthermore, both the wild-type strain and *dinB* knockout gained a mutation in an intergenic region near *ampC*. We also identified a mutation in *rpoD*, the gene encoding sigma factor 70, in two of the *katE* replicates, as well as in two of the *dinB* and one of the MG1655 replicates. Despite this mutation not meeting our primary criteria of 60% frequency and occurrence in at least two replicates, we found it worth investigating due to its remarkable consistency and recurrence across different strains, all showing an identical mutation. This suggests an important role for *rpoD* in the adaptation process.

## A transposon region containing *ampC* is amplified by amoxicillin exposure

Gene copy analysis revealed that all three strains amplified the same collection of genes after adaptation to amoxicillin (Fig 2), including the *frd* cluster, *ampC*, *blc*, *gdx*, *ecnA*, *ecnB*, *efp*, *epmA*, *epmB*, and a gene coding for the uncharacterized protein YjeJ, some of them by more than 10 additional copies (Fig 2). Parts of this region have been previously described as a potential transposon region and are transferable between *E. coli* cells [49]. We additionally observed unique gene amplifications in each strain. In the wild-type strain MG1655, unique amplifications were identified in *mscM* and *yjeO*. The *katE* knockout strain showed unique amplifications in *frsA*, and *nhaR*. The strain also amplified various transposable elements, namely *insA6*, *insB1*, *insB5*, *insB6*, and *insB9*. The *dinB* knockout strain exhibited unique amplifications in *aspA*, *cutA*, *dcuA*, *dsbD*, *fxsA*, *groL*, *groS*, *insG*, *psd*, *yjdC*, *yjeH*, and *yjeI*.

## Transcriptomic Changes in Knockout Mutants highlight potential compensatory Mechanisms

The deletion of the *dinB* and *katE* genes resulted in distinct transcriptomic changes compared to the wild type (Fig 3). In the Δ*dinB* mutant, genes associated with error-free translesion synthesis were significantly downregulated, as well as genes involved in error-prone translesion synthesis. The Δ*katE* mutant exhibited a downregulation of hydrogen peroxide catabolic and

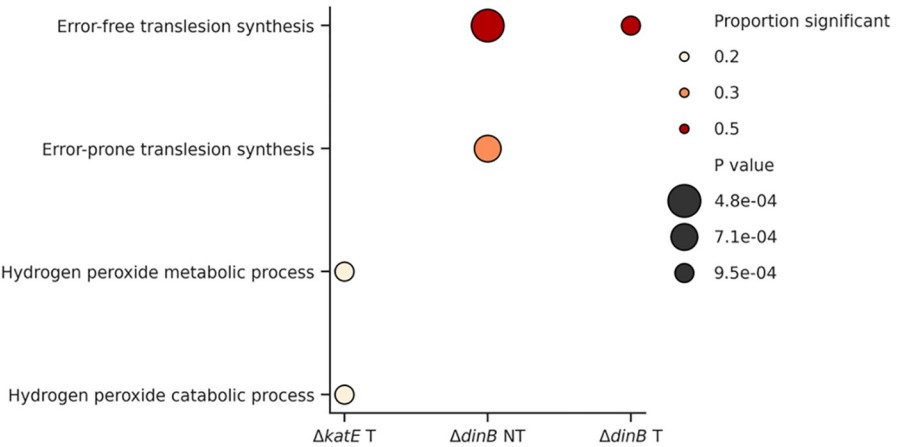

**Fig 3. Transcriptomic changes in knockout mutants—Downregulation.** The figure illustrates the proportion of significant transcriptomic changes across the knockout mutants. The size of the data points relates to the proportion of significant changes, whereas the colour code indicates p-values. NT stands for non-treated, while T means treated with ¼ MIC during overnight incubation and before RNA isolation.

metabolic processes, but this effect was less pronounced than the impact observed on pathways related to the Δ*dinB* knockout. This suggests that while *katE* is involved in managing oxidative stress, other pathways can partially compensate for its absence, whereas the loss of *dinB* appears to be less easily compensated.

## Influence of adaptation and mutations on the transcriptome

All three strains—wild type (MG1655), Δ*dinB*, and Δ*katE*—demonstrated a significant upregulation of *ampC* at the end of the evolution experiment (Fig 4). In the wild-type MG1655, the most significant downregulation in the transcriptome could be found in *waaB* (LPS synthesis, involved in host invasion [50]), *asnA* (asparagine synthetase), and *gstB* (oxidative stress), alongside the significant upregulation of *mdtK* (multidrug efflux pump) and *yhaV* (toxin of the YhaV-PrlF toxin-antitoxin system).

The Δ*dinB* strain showed significant downregulation of *alaE* (alanine exporter), *tatE* (transport of folded proteins, virulence factor [51]), and *hdeA* (acid stress response), with upregulation of *panB* (pantothenate biosynthesis pathway, involved in resistance in *Edwardsiella tarda* [52]) and *efp* (elongation factor, peptide bond synthesis). The Δ*katE* strain displayed

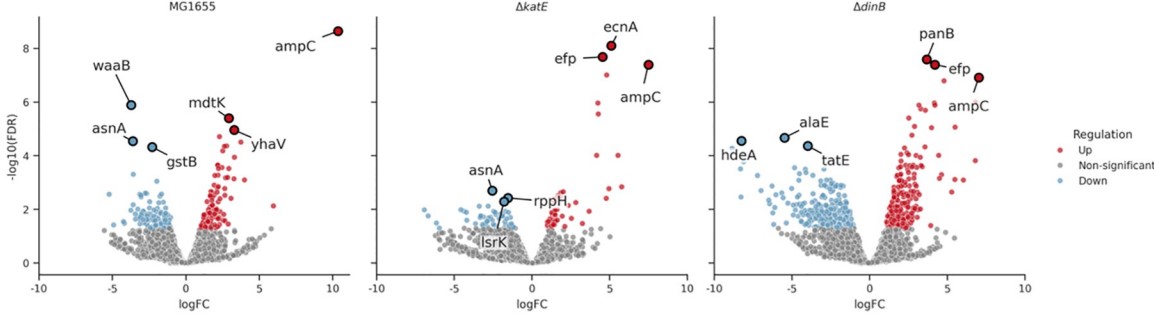

**Fig 4. Top three up-and down-regulated genes after AMO adaptation.** Blue dots illustrate significant downregulation, red dots significant upregulation.

downregulation of *asnA*, *lsrK* (quorum sensing), and *rppH* (mRNA degradation), and upregulation of *efp* and *ecnA* (antidote, programmed cell death).

Across all strains, there was a general trend of more upregulated pathways than downregulated during amoxicillin adaptation. The Δ*dinB* strain exhibited the most extensive changes in pathway regulation, indicating a significant transcriptomic shift in response to both, the gene knockout and the amoxicillin exposure.

## Evolved Knockout strains both downregulated *prlF* and *yhaV*, but upregulated distinct sets of genes

In the Δ*dinB* strain, the transcriptomic data revealed a significant downregulation of *prlF* and *yhaV* compared to the evolved wild-type. PrlF and YhaV are known to be a toxin-antitoxin system in *E. coli* [53]. The Δ*dinB* strain furthermore exhibited the most significant upregulation of *fxsA* and *yjeI*. The *fxsA* gene is implicated in stress response pathways [54], while *yjeI* is a Ser/Thr kinase [55].

The Δ*katE* strain demonstrated a different set of genes that were the most significantly upregulated, including *ecnA* and *efp*. The e*cnA* gene is part of the entericidin operon, involved in bacterial programmed cell death and stress responses [56]. The *efp* gene encodes an elongation factor and mutations in the gene have been associated with suppressed lethality of a *rep/uvrD* double mutant [57].

Overall, comparing the evolved knockout strains to the evolved wild-type disclosed that both Δ*dinB* and Δ*katE* mutants adapted by downregulating the common stress response regulators *prlF* and *yhaV*. Despite this shared downregulation, each strain utilized distinct upregulated pathways. The Δ*dinB* strain strongly focused on upregulating genes related to cell envelope stability and stress tolerance, while the Δ*katE* strain mostly adapted expression of genes associated with stress response and iron metabolism.

## RpoD mutation impacts protein structure

To better understand the potential impact of the mutation in *rpoD* on the protein structure, we compared the predicted structures of the mutated protein with the wild-type protein. The expected stability change of the protein after the mutation (D445V) was 0.06 kcal/mol (stabilizing). Based on the model, the mutation D445V causes the protein to lose numerous bonds to the neighbouring amino acids (Fig 5). Notably, the D445V mutation occurs within region 2 of

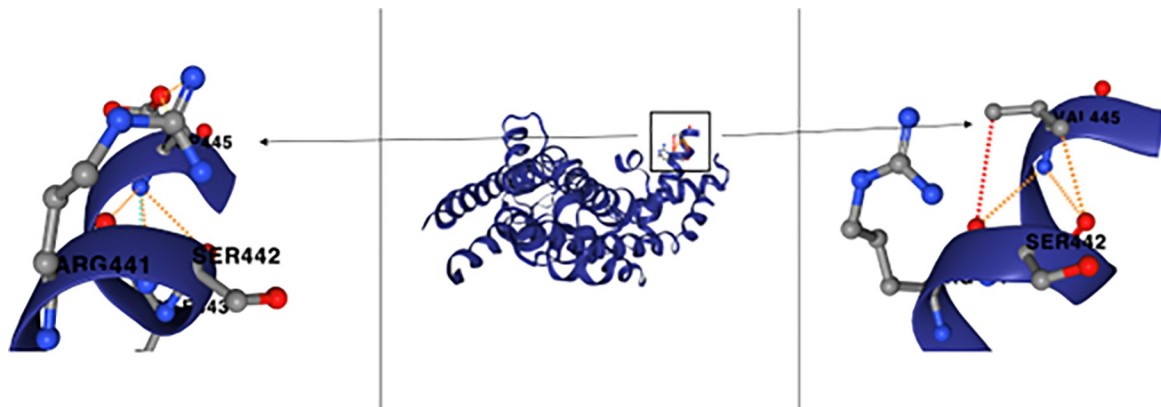

**Fig 5. RpoD protein prediction.** Wild-Type D445 left, Mutant V445 right. Interactions are illustrated with the following colour code. Ionic (yellow), polar (orange), hydrogen bond (red), VDW (light blue).

the sigma 70 factor, a highly conserved region critical for both DNA binding (specifically, -10 promoter recognition) and RNA polymerase (RNAP) binding. The mutation could potentially affect both RNAP binding and promoter recognition, which are essential for the transcription of numerous genes.

## 4. Discussion

The generally assumed roles of *dinB* and *katE* in beta-lactam resistance mechanisms in *E. coli* need to be reconsidered. Contrary to previous assumptions, the adaptation rates to amoxicillin in the absence of *dinB* and *katE*, were similar to those observed in wild-type *E. coli*. The uniformity observed in the adaptation rates of both wild-type and knockout strains indicates shared adaptation mechanisms or the activation of compensatory pathways within the bacterial genome. This collective adaptation hypothesis gains support from the limited set of mutations shared across strains and the consistent pattern of gene amplifications observed in a region previously identified as a 'pre-plasmid' [49]. Notably, this region encompasses the *ampC* gene and its corresponding operon. Amplification of the *ampC* gene serves as a survival mechanism in cells exposed to beta-lactams [58]. However, amplification alone does not confer high resistance levels unless accompanied by specific mutations [47,58,59]. Most described in clinical isolates are mutations changing the -35 and -10 box, and mutations in the attenuator region (+17 to +37) [60–63], which corresponds to our findings.

Some mutations in *frdD* were detected in at least two of the three biological replicates, suggesting a potential role in adaptation. The *frd* operon, which encodes the fumarate reductase enzyme, overlaps with the *ampC* promoter, and its terminator acts as an attenuator for *ampC* expression [45]. Mutations in the *frd* operon have been described to boost *ampC* amplification and contribute to beta-lactam resistance [47,59]. However, fumarate itself seems to be involved in persister formation, as cells carrying a plasmid containing both *ampC* and the *frd* operon show a higher resistance to beta-lactams than cells containing only *ampC* [64]. Additionally, cells lacking *ampC* still adapt to high concentrations of beta-lactam antibiotics like ampicillin, indicating the presence of alternative adaptation mechanisms that do not rely on *ampC* amplification [65].

Furthermore, our data show that the *frdD* gene was constantly upregulated in response to amoxicillin exposure, even in strains that had already adapted to the antibiotic. In contrast, *ampC* transcription remained unchanged upon amoxicillin exposure in these adapted cells. This suggests a more complex role of fumarate in amoxicillin adaptation beyond *ampC* regulation. The *frdD* gene is integral to anaerobic respiration in bacteria, and the efficacy of beta-lactam antibiotics is linked to bacterial respiration [66,67]. Therefore, changes in anaerobic respiration patterns mediated by *frdD* expression may affect bacterial fitness and susceptibility to antibiotics. Moreover, *frdD* also plays a role in energy metabolism. Beta-lactam antibiotics primarily target cell wall synthesis by inhibiting the activity of penicillin-binding proteins (PBPs). This inhibition has been associated with changes in TCA cycle activity and oxidative phosphorylation [19,68]. Consequently, changes in energy availability and cellular redox status, influenced by *frdD* expression, may affect the efficiency of cell wall synthesis, and by extension, bacterial susceptibility to beta-lactam antibiotics.

Additionally, *frdD* could be directly involved in beta-lactam adaptation as mediator of metabolic flexibility under stress conditions. Fumarate serves as an alternative electron acceptor under anaerobic conditions and is reduced to succinate by fumarate reductase [69]. In certain conditions, fumarate reductase can substitute for succinate dehydrogenase, a component of complex II in aerobic respiration that also participates in the Krebs cycle [70,71]. This

substitution may reflect a metabolic adaptation to different growth conditions, as fumarate reductase is associated with higher superoxide anion production than succinate dehydrogenase [70,72].

Although producing more superoxide anions might seem counterintuitive during oxidative stress, it can be beneficial for two reasons. First, superoxide production can aid as a signal to activate stress response pathways, helping cells adapt to environmental changes. Superoxide is metabolized into oxygen and hydrogen peroxide by superoxide dismutase, and hydrogen peroxide is subsequently transformed by catalases and peroxidases [73]. Superoxide dismutase activity has been linked to antibiotic resistance by activating the stringent response, upregulating efflux pumps, downregulating of the outer membrane porin OmpF, and co-regulation of the multidrug-resistant locus mar [73–76]. Second, elevated superoxide levels can increase mutagenesis, which, while potentially harmful, also generates genetic diversity [76,77]. This diversity can be advantageous under selective pressures, providing a collection of mutations from which beneficial traits, like antibiotic resistance, can emerge.

The recurrent *rpoD* mutation identified in both knockout and wild-type strains suggests a key role for *rpoD* in the adaptive response to amoxicillin. Although the precise functioning of this mutated sigma factor under antibiotic stress is presently unclear, the consistent appearance of identical mutations across various strains points to a significant regulatory function. Being a highly conserved housekeeping gene across bacteria [78], *rpoD* modulates 60–95% of sigma factors during exponential growth and binds to over half of all sigma factor binding sites across the genome [79–81]. Mutants of *rpoD* generated through site-directed mutagenesis, exhibit genome-wide transcriptomic changes that lead to phenotypes with improved stress tolerance [82]. While mutating *rpoD* seems disadvantageous due to a likely fitness loss and a high chance of lethality, MAGE-seq measurements reveal that the specific D445V mutation has a limited impact on viability [83,84]. Surprisingly, this mutation occurs in one of the most conserved areas of the gene, region 2, which binds to the -10 motif and unwinds the DNA duplex [85]. However, such mutations in highly conserved regions of sigma factors have been noted in other studies on antibiotic resistance and long-term adaptation [86].

The D445V mutation has been highlighted in recent literature, in a study involving a Chron's disease-associated *E. coli* strain [87]. It was shown not to affect the catalytic activity of RNA polymerase (RNAP), yet it influences genes related to gut colonization and beta-lactam resistance. The study found transcriptomic changes in key genes, including *ampC*, the ROS responder *yggE*, and the *rpoS* regulator *rprA*. Notably, D445V was sufficient to render the previously susceptible strain resistant to five beta-lactams and increase resistance to other tested antibiotics such as ciprofloxacin [87].

Given the D445V mutation's position in region 2 and the predicted changes to the protein structure, it seems likely that this mutation affects the sigma factor's ability to bind to RNA polymerase. RpoD and RpoS (Sigma 30) share several binding sites [88] and compete for RNAP [89,90]. The D445V mutation might enhance RpoD's affinity to bind to RNAP, giving it an advantage over RpoS. Indeed, the previously described D445V mutant showed lower levels of sigma S [87]. While it may seem counterintuitive to inhibit RpoS, the sigma factor responsible for the general stress response in *E. coli* [91,92], especially due to its involvement in antibiotic-induced mutagenesis [16,93], this strategy might favour bacterial survival under specific conditions where maintaining fitness outweighs the need for broad stress response, making the attenuation or loss of RpoS advantageous.

One replicate of MG1655 and one replicate of Δ*dinB* additionally carried an identical *prlF* mutation (S2 Table). Nevertheless, MG1655 upregulated the toxin-antitoxin system (TA), and the *dinB* knockout downregulated it upon adaptation and exposure to amoxicillin. The mutation we observed in *prlF* causes downregulation of *ompF*, thus leading to resistance to multiple

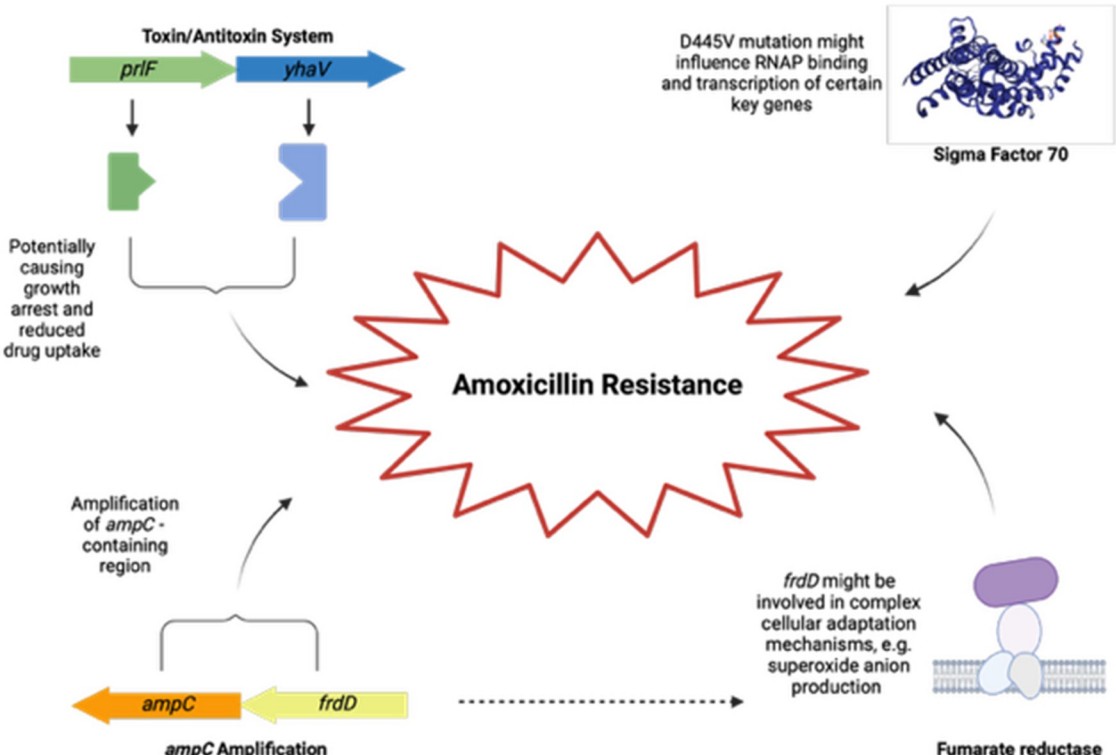

**Fig 6. Conclusion.** Key genetic factors in amoxicillin resistance acquisition in E. coli. Created in BioRender.com.

antibiotics, i.e. carbenicillin, and aztreonam [94]. OmpF is involved in beta-lactam uptake, and mutations and porin loss are known to be involved in beta-lactam resistance [95,96]. How the mutation in *prlF* changes its interaction with YhaV is unknown. YhaV does not only cleave cellular mRNAs like *ompF* [97], but it also causes reversible bacteriostasis that is neutralized by PrlF [53,98]. Bacteriostasis could indirectly impact amoxicillin efficacy. Beta-lactams are most effective against actively growing and dividing bacteria [99]. Thus, through growth arrest, the cell might counteract the antibiotic's primary target–penicillin-binding proteins involved in the final stages of peptidoglycan cross-linking [100].

Our findings give rise to several hypotheses (Fig 6). First, the consistent mutations in *frdD* and the *ampC* operon suggest that the regulatory role of *frdD* on *ampC* is crucial for beta-lactam resistance, but fumarate's role in this process is likely several pathways. Furthermore, the recurrent *rpoD* mutations indicate a significant regulatory role for this sigma factor under antibiotic stress. Third, the involvement of the TA system *prlF/yhaV* suggests that this system might have a role in the acquisition of antimicrobial resistance which is not very well understood in the framework of the presently available knowledge. Lastly, our data set shows that despite being limited in stress responses, the *dinB* and *katE* knockout strains had high adaptability, enabling a seemingly unhindered acquisition of amoxicillin resistance.

## Supporting information

**S1 Fig. Frequency distributions of mutations.** The graph illustrates the frequency distributions of mutations across the samples.
(TIF)

**S1 Table. PrlF mutation.**
(DOCX)

**S2 Table. Antibiotic concentrations RNA extraction.**
(DOCX)

## Acknowledgments

The authors thank Stanley Brul and Meike Wortel for stimulating discussions.

## Author Contributions

**Conceptualization:** Lisa Teichmann, Johan Bengtsson-Palme, Benno ter Kuile.

**Data curation:** Lisa Teichmann, Marcus Wenne, Gaurav Dugar.

**Formal analysis:** Lisa Teichmann, Marcus Wenne, Gaurav Dugar, Johan Bengtsson-Palme.

**Funding acquisition:** Benno ter Kuile.

**Investigation:** Lisa Teichmann, Sam Luitwieler, Benno ter Kuile.

**Methodology:** Lisa Teichmann, Sam Luitwieler, Benno ter Kuile.

**Software:** Marcus Wenne.

**Supervision:** Benno ter Kuile.

**Writing – original draft:** Lisa Teichmann.

**Writing – review & editing:** Marcus Wenne, Sam Luitwieler, Gaurav Dugar, Johan Bengtsson-Palme, Benno ter Kuile.

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
