## [Decision Letter · Decision Letter 0]

28 Nov 2024

PONE-D-24-43795Genetic Adaptation to Amoxicillin in Escherichia coli: The Limited Role of dinB and katEPLOS ONE

Dear Dr. Kuile,

Thank you for submitting your manuscript to PLOS ONE. After careful consideration, we feel that it has merit but does not fully meet PLOS ONE’s publication criteria as it currently stands. Therefore, we invite you to submit a revised version of the manuscript that addresses the points raised during the review process.

We look forward to receiving your revised manuscript.

Kind regards,

Bashir Sajo Mienda, PhD

Academic Editor

PLOS ONE

Journal Requirements:

“This study was financed by The Netherlands Food and Consumer Product Safety Authority”

“This research was financed by the Netherlands Food and Consumer Product safety Authority. The authors thank Stanley Brul and Meike Wortel for stimulating discussions. “

“This study was financed by The Netherlands Food and Consumer Product Safety Authority”

Reviewers' comments:

Reviewer's Responses to Questions

**Comments to the Author**

1. Is the manuscript technically sound, and do the data support the conclusions?

Reviewer #1: Yes

2. Has the statistical analysis been performed appropriately and rigorously? 

Reviewer #1: N/A

3. Have the authors made all data underlying the findings in their manuscript fully available?

Reviewer #1: Yes

4. Is the manuscript presented in an intelligible fashion and written in standard English?

Reviewer #1: Yes

5. Review Comments to the Author

Reviewer #1: Dear Author,

Thank you for your effort to share your research findings with the scientific community through this manuscript entitled, “Genetic Adaptation to Amoxicillin in Escherichia coli: The Limited Role of dinB and katE” and submitting it for publication in PLOS ONE. I have gone through the manuscript draft and found it quite interesting. The manuscript data is technically sound and supports the conclusions drawn. The description of the experiments conducted indicated that it was rigorously carried out, with appropriate controls, replication, and sample sizes. There is adequate evidence that shows data underlying the findings in manuscript are made fully available and presented in an intelligible fashion and written in standard English. The paper is clear and well-composed. The dataset therein is very useful. There is sufficient justification to affirm the originality of the work, numerical figures of the various analytical factors have been reported with a reasonable degree of accuracy. The objectives of the work as outlined in the introductory part of the manuscript have been reasonably achieved. Thus, the work could be adjudged complete within its limitations. In a nutshell, this is a high-quality manuscript that clarified the contributions of DNA polymerase IV (dinB) and catalase HP2 (katE) genes in E. coli's adaptation to beta-lactam resistance. In particular, the work has established that dinB and katE may not be as critical for beta-lactam adaptation as generally assumed. This could be quite thought-provoking to the scientific community However; I have outlined some comments that could help improve the quality of the manuscript.

Comments:

1. Line 155. There may be no need of a hyphen between 5 and minute, also the word ‘minutes’ should be written in the plural form, to make it consistent with the other usage elsewhere in the manuscript.

2. L192. The concentration of the antibiotic used needs to be specified.

3. L196. Specify the concentration of the agarose used

4. L439 - 440 The direct involvement of frdD should be further elaborated to adequately discuss its role based on the findings of this study.

6. PLOS authors have the option to publish the peer review history of their article (what does this mean?). If published, this will include your full peer review and any attached files.

Reviewer #1: No

---

## [Author Response · Author response to Decision Letter 0]

10 Dec 2024

Response to the reviewer PONE-D-24-43795

Teichmann et al.,: Genetic Adaptation to Amoxicillin in Escherichia coli: The Limited Role of dinB and katE

Reviewers comments: 

Thank you for providing me the opportunity to review this manuscript titled, “Genetic Adaptation to Amoxicillin in Escherichia coli: The Limited Role of dinB and katE” for publication in your journal (PLOS ONE). I have gone through the manuscript draft and found it quite interesting. The paper is clear and well-composed. The dataset therein is very useful. There is sufficient justification to affirm the originality of the work, numerical figures of the various analytical factors have been reported with a reasonable degree of accuracy. The objectives of the work as outlined in the introductory part of the manuscript have been reasonably achieved. Thus, the work could be adjudged complete within its limitations. In a nutshell, this is a high-quality manuscript that clarified the contributions of DNA polymerase IV (dinB) and catalase HP2 (katE) genes in E. coli's adaptation to beta-lactam resistance. In particular, the work has established that dinB and katE may not be as critical for beta-lactam adaptation as generally assumed. This could be quite thought-provoking to the scientific community However; I have outlined some comments that could help improve the quality of the manuscript. 

Comments: 

1. Line 155. There may be no need of a hyphen between 5 and minute, also the word ‘minutes’ should be written in the plural form, to make it consistent with the other usage elsewhere in the manuscript. 

Accepted and corrected accordingly

2. L192. The concentration of the antibiotic used needs to be specified. 

Accepted and corrected accordingly

3. L196. Specify the concentration of the agarose used 

Accepted and corrected accordingly

4. L439 - 440 The direct involvement of frdD should be further elaborated to adequately discuss its role based on the findings of this study. 

Frankly, we found this comments slightly confusing. The Involvement of frdD is extensively discussed in line 414-46. The combined authors came up with the following ideas to address this comment: 

1. "Mutations in the frd operon have been described to boost ampC amplification and contribute to beta-lactam resistance”: Here, I guess we could compare the expression of ampC in strains with and without these mutations to add to the data in the cited papers

These data turned out very hard to compare, confusing instead of clarifying the issue. 

2. "changes in energy availability and cellular redox status, influenced by frdD expression, may affect the efficiency of cell wall synthesis, and by extension, bacterial susceptibility to beta-lactam antibiotics.”: This is a long-shot, but could we look at expression of e.g. cell wall synthesis pathway genes in relation to the frdD expression? If we see an association, I guess that would strengthen this case, but this might be hard as I know energy metabolism is affected by a bunch of different things.

This association didn’t materialize either.

3. We already analysed the fumarate reductase/succinate dehydrogenase relationship in detail, so this cannot be expanded.

4. "Superoxide dismutase activity has been linked to antibiotic resistance by activating the stringent response, upregulating efflux pumps, downregulating of the outer membrane porin OmpF, and co-regulation of the multidrug-resistant locus mar”: Do we see these upregulations together with increased frdD levels in our data?

Turns out that this is not the case either. 

5. "elevated superoxide levels can increase mutagenesis”: We have the transcriptomics data from the final strains. It might be interesting to compare the frdD levels at the mid-point to the number of mutations in the corresponding final evolved strains. However, this may be comparing apples to oranges. 

Indeed, when this analysis was tried, the outcome was an uninterpretable mess. 

In short, since the involvement of frdD has already been discussed extensively in the framework oh the literature (lines 414-46), we have decided not to add to that. We hope to have convinced you that this is not out of disrespect for the reviewer or lack of will to make the effort.

---

## [Decision Letter · Decision Letter 1]

29 Dec 2024

Genetic Adaptation to Amoxicillin in Escherichia coli: The Limited Role of dinB and katE

PONE-D-24-43795R1

Dear Dr. KUILE,

We’re pleased to inform you that your manuscript has been judged scientifically suitable for publication and will be formally accepted for publication once it meets all outstanding technical requirements.

Kind regards,

Bashir Sajo Mienda, PhD

Academic Editor

PLOS ONE

Additional Editor Comments (optional):

Reviewers' comments:

Reviewer's Responses to Questions

**Comments to the Author**

1. If the authors have adequately addressed your comments raised in a previous round of review and you feel that this manuscript is now acceptable for publication, you may indicate that here to bypass the “Comments to the Author” section, enter your conflict of interest statement in the “Confidential to Editor” section, and submit your "Accept" recommendation.

Reviewer #1: All comments have been addressed

2. Is the manuscript technically sound, and do the data support the conclusions?

Reviewer #1: Yes

3. Has the statistical analysis been performed appropriately and rigorously? 

Reviewer #1: I Don't Know

4. Have the authors made all data underlying the findings in their manuscript fully available?

Reviewer #1: Yes

5. Is the manuscript presented in an intelligible fashion and written in standard English?

Reviewer #1: Yes

6. Review Comments to the Author

Reviewer #1: (No Response)

7. PLOS authors have the option to publish the peer review history of their article (what does this mean?). If published, this will include your full peer review and any attached files.

Reviewer #1: No

---

## [Editor Report · Acceptance letter]

7 Jan 2025

PONE-D-24-43795R1 

PLOS ONE

Dear Dr. ter Kuile, 

I'm pleased to inform you that your manuscript has been deemed suitable for publication in PLOS ONE. Congratulations! Your manuscript is now being handed over to our production team.

Kind regards, 

on behalf of

Dr. Bashir Sajo Mienda 

Academic Editor

PLOS ONE